# A Unified and Data-Efficient Framework for Out-of-Distribution and Generalization

## Abstract

Machine learning classification models inevitably encounter two types of out-of-distribution (OOD) data in practical applications, i.e., covariate-shifted data and semantic-shifted data. Current approaches typically address them separately in OOD detection and generalization tasks, resulting in limited capability to handle both types of OOD data simultaneously. This limitation motivates us to tackle both challenges within a unified framework. However, our theoretical investigation uncovers a conflict in jointly addressing these two problems, namely *Optimization Conflict* (OC). Moreover, collecting OOD data remains a significant challenge, making it difficult to provide models with sufficient OOD samples for effective learning. To this end, we propose a novel method called *Tackling OOD Detection and Generalization* (TODG), which incorporates a regularization term to mitigate OC and employs a data generation strategy to alleviate the scarcity of OOD data. Extensive experiments demonstrate that TODG outperforms existing methods, showcasing its effectiveness in both OOD detection and generalization tasks.

## 1 Introduction

Machine learning classification models deployed in real-world scenarios often encounter *out-of-distribution* (OOD) data that deviates from the in-distribution (ID) training data, including both *covariate-shifted data*, i.e., data residing within the same semantic space as ID data (Wang et al., 2023a) and *semantic-shifted data*, i.e., data originating from novel, unknown classes with shifted semantic space from ID data (Wang et al., 2025). Towards these two kinds of OOD data, recent research efforts have focused on two key directions: OOD detection (Du et al., 2022; Ming et al., 2022), i.e., detecting whether encountered data exhibits the *semantic-shift property*, and OOD generalization (Ye et al., 2022; Zhou et al., 2023; Liao et al., 2024), i.e., improves the generalization of model performance against *covariate-shifted data*. Given their divergent goals, these two research domains have progressed independently along parallel trajectories (Yang et al., Jun. 2024).

However, realistic applications often tends to encounter **mixed** data structure including both *covariate-shifted data* and *semantic-shifted data*. For instance, a classifier, well-trained in a grassland environment, may encounter covariate-shifted inputs (e.g., the wolf in the middle of Fig. 1) and semantic-shifted inputs (e.g., the tiger on the right side of Fig. 1) in a snowy environment. Consequently, a natural question arises to achieve an unified framework accounting for both detection and generalization towards such mixed OOD data structure:

> *Can we develop a unified approach to jointly address OOD detection and OOD generalization for handling both covariate and semantic shifted OOD data?*

To this end, we conduct theoretical analysis to inform a fundamental challenge to optimize **both** the objectives of OOD detection and OOD generalization simultaneously. To be specific, we introduce a novel feature disentanglement framework (see Fig. 1) based on our theoretical insight that data within the same class of identical distribution have a bounded feature distribution (see Theorem 2). By decomposing the OOD data into class-oriented and environment-related components, our theory shows an *Optimization Conflict* (OC) that environment-related features, i.e., class-irrelevant components that should be suppressed, receive anti-intuitive high scores in covariate-shifted data during

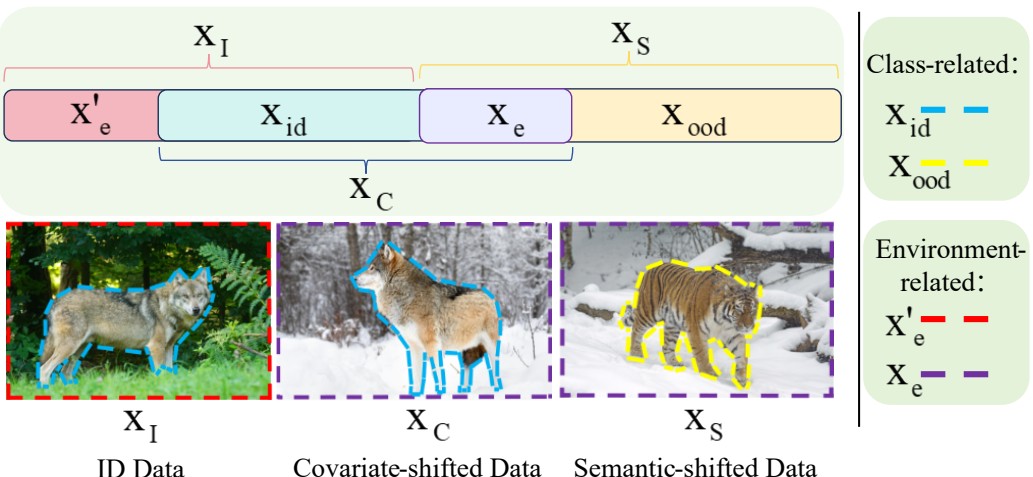

Figure 1: The feature representation model. It characterizes the relationships among ID data ($\mathbf{x}_I$), covariate-shifted OOD data ($\mathbf{x}_C$), and semantical-shifted OOD data ($\mathbf{x}_S$).

optimization (cf. Theorem 1). In addition, the complexity of open scenarios and the unpredictability of OOD data pose additional challenges in collecting enough OOD data for training.

To overcome the above two challenges, we propose a novel method named *Tackling OOD Detection and Generalization* (TODG), which eliminates OC and augments the insufficient OOD data simultaneously. To be specific, TODG equips a novel risk regularization mechanism with theoretical support (cf. Theorem 3) to effectively address the conflict phenomenon during optimization of both OOD detection and generalization tasks. Meanwhile, to address the data-insufficiency challenge, TODG employs an implicit data generation strategy, which estimates the mean and variance of the limited available OOD data and samples from the corresponding Gaussian distributions.

We summarize the main contributions of this paper as follows.

- Theoretically, we analyze the inherent *Optimization Conflict* (OC) when optimizing both OOD detection and generalization tasks simultaneously, and rigorously prove that our proposed regularization method effectively mitigates this critical issue.

- Techically, we propose an novel algorithm that simultaneously tackles OOD detection and generalization while resolving the underlying OC. To the best of our knowledge, our work is the first to analyze and address the OC when simultaneously handling the two tasks.

- Empirically, we conduct extensive experiments on both OOD detection and generalization tasks, comprehensively demonstrating that TODG outperforms previous state-of-the-art (SOTA) methods in the respective domains, further validating its effectiveness.

## 2 PRELIMINARY

In this section, we define the problem and analyze the inherent OC in jointly tackling OOD detection and generalization in the manner of Outlier Exposure (OE) (Hendrycks et al., 2019).

### 2.1 NOTATIONS AND PROBLEM SETUP

Let $\mathcal{X}$ represent the input space and $\mathcal{Y} = \{1, \cdots, c\}$ represent the ID label space. Define $X_I$ and $Y_I$ as random variables with outputs from $\mathcal{X}$ and $\mathcal{Y}$, respectively. The ID joint distribution over $\mathcal{X} \times \mathcal{Y}$ is denoted by $D_{X_I Y_I}$ (Fang et al., 2022). Additionally, $D_{X_O^{cov} Y_I}$ represents the covariate-shifted OOD joint distribution, while $D_{X_O^{sem} Y_O}$ denotes the semantic-shifted OOD joint distribution, where $X_O^{cov}$ and $X_O^{sem}$ are random variables from $\mathcal{X}$, and $Y_O$ is an unknown random variable with values outside $\mathcal{Y}$, i.e., $Y_O \notin \mathcal{Y}$. Besides, for the convenience, we use $\mathtt{sof}$ to refer the "softmax" operation when

computing the cross-entropy loss. During testing, we encounter the mixed distribution:

$$D_{XY} := \lambda^{cov} D_{X_O^{cov} Y_I} + \lambda^{sem} D_{X_O^{sem} Y_O} + (1 - \lambda^{cov} - \lambda^{sem}) D_{X_I Y_I}, \tag{1}$$

where the constants $\lambda^{cov}$ and $\lambda^{sem}$ represent unknown class-prior probabilities, with $\lambda^{cov}, \lambda^{sem}, \lambda^{cov} + \lambda^{sem} \in [0, 1)$. However, we can only observe the marginal distribution:

$$D_X := (1 - \lambda^{cov} - \lambda^{sem}) D_{X_I} + \lambda^{cov} D_{X_O^{cov}} + \lambda^{sem} D_{X_O^{sem}}. \tag{2}$$

We also have the ID training data defined as follows:

$$S_I := \{(\mathbf{x}_I^1, y_I), (\mathbf{x}_I^2, y_I), \cdots, (\mathbf{x}_I^n, y_I)\} \overset{\text{i.i.d.}}{\sim} D_{X_I Y_I}. \tag{3}$$

We consider the well-trained classification model $\mathbf{f}(\cdot; \mathbf{w}) : \mathcal{X} \mapsto \mathbb{R}^c$ with the logit outputs, parameterized by $\mathbf{w}$ from the parameter space $\mathcal{W}$. To address both OOD detection and generalization tasks simultaneously, we formally define the problem as follows.

**Problem 1.** *Given the ID training data $S_I$ in Eq. (3), the objective of the joint optimization task for OOD detection and generalization is to train a classifier $\mathbf{f}$ based on $S_I$ such that, for any test data $\mathbf{x}$ drawn from the mixed marginal distribution $D_X$: (a) if $\mathbf{x}$ is an observation from distribution $D_O^{sem}$, $\mathbf{f}$ can identify $\mathbf{x}$ as semantic-shifted OOD data; and (b) if $\mathbf{x}$ is an obervation from distribution $D_{X_I}$ or $D_{X_O^{cov}}$, $\mathbf{f}$ can classify $\mathbf{x}$ into its correct ID class.*

**Remark 1.** *Compared to standard OOD detection tasks, Problem 1 further considers the classification of covariate-shifted OOD data, making it more aligned with real-world application scenarios and significantly more challenging.*

## 2.2 DETECTING SEMANTIC-SHIFTED OOD DATA

We commence by focusing on the first task to be addressed in Problem 1, i.e., detecting OOD data. In the field of OOD detection, methods typically address this by employing an OOD scoring function. Specifically, these approaches leverage a pre-trained classifier equipped with the OOD scoring function, denoted as $s(; \mathbf{f}) : \mathcal{X} \mapsto \mathbb{R}$, which assigns higher scores to ID data and lower scores to OOD samples (Hendrycks & Gimpel, 2017; Wang et al., 2023b). The final ID/OOD distinction is achieved via level-set estimation of the obtained scores, formalized as:

$$h(\mathbf{x}) = \text{ID, if } s(\mathbf{x}; \mathbf{f}) \geq \rho; \text{ otherwise } h(\mathbf{x}) = \text{OOD}, \tag{4}$$

where $h(\mathbf{x})$ is the binary decision between ID and OOD and $\rho$ is a pre-defined parameter.

**Outlier Exposure (OE) Manner.** Despite the emergence of many advanced OOD scoring functions, satisfactory OOD detection performance remains unattained due to the lack of knowledge about OOD data. To this end, Hendrycks et al. (2019) proposed Outlier Exposure (OE), which fine-tunes the model using additional auxiliary OOD data $S_O^{sem}$ to enhance the model ability to distinguish between ID and semantic-shifted OOD data, where $S_O^{sem}$ can be formulated in the following.

$$S_O^{sem} := \{\mathbf{x}_S^1, \mathbf{x}_S^2, \cdots, \mathbf{x}_S^m\} \overset{\text{i.i.d.}}{\sim} D_{X_O^{sem}}. \tag{5}$$

We provide the formal definition of OE as follows.

**Definition 1** (Outlier Exposure (OE)). *(Hendrycks et al., 2019) Given the ID training data $S_I$ and the auxiliary OOD data $S_O^{sem}$, the objective of OE is to train the model $\mathbf{f}$ on such data such that it excels at OOD detection. Specifically, for any input $\mathbf{x}$: (a) if $\mathbf{x}$ is an observation from $D_{X_O^{sem}}$, $\mathbf{f}$ can detect $\mathbf{x}$ as an OOD case; and (b) if $\mathbf{x}$ is an observation from $D_{X_I}$, $\mathbf{f}$ can classify $\mathbf{x}$ into its correct ID class.*

**Remark 2.** *The classic OE, as a work in the field of OOD detection, does not specifically address the problem of OOD generalization. Building upon the idea of OE, our work leverages OOD data with covariate shift to jointly train the model, thereby further enhancing its performance in OOD generalization tasks, i.e., improving its ability to correctly classify samples from distribution $D_{X_O^{cov}}$.*

**Risk formulation of OE.** OE achieves its objective by minimizing the empirical risk:

$$\widehat{R}(\mathbf{f}) = \widehat{R}_I(\mathbf{f}) + \alpha \cdot \widehat{R}_O^{sem}(\mathbf{f}), \tag{6}$$

where $\alpha$ is a trade-off parameter. The first term addresses the classification of ID data while increasing their OOD scores. The second term deals with semantic-shifted OOD data, aiming to reduce their OOD scores. We also provide the expected form of Eq. (6) as

$$R(\mathbf{f}) = R_{\mathrm{I}}(\mathbf{f}) + \alpha \cdot R_{\mathrm{O}}^{sem}(\mathbf{f}). \tag{7}$$

Specifically, $R_{\mathrm{I}}(\mathbf{f})$ and $R_{\mathrm{O}}^{\mathrm{sem}}(\mathbf{f})$ can be difined as:

$$R_{\mathrm{I}}(\mathbf{f}) = \mathbb{E}_{(\mathbf{x}_{\mathrm{I}}, y) \sim D_{X_{\mathrm{I}} Y_{\mathrm{I}}}} \ell_{\mathrm{I}}(\mathbf{f}(\mathbf{x}_{\mathrm{I}}; \mathbf{w}), y) \text{ and } R_{\mathrm{O}}^{\mathrm{sem}}(\mathbf{f}) = \mathbb{E}_{\mathbf{x}_{\mathrm{S}} \sim D_{X_{\mathrm{O}}^{sem}}} \ell_{\mathrm{S}}(\mathbf{f}(\mathbf{x}_{\mathrm{S}}; \mathbf{w})). \tag{8}$$

Similarly, $\widehat{R}_{\mathrm{I}}(\mathbf{f})$ and $\widehat{R}_{\mathrm{O}}^{sem}(\mathbf{f})$ can be difined as:

$$\widehat{R}_{\mathrm{I}}(\mathbf{f}) = \frac{1}{n} \sum_{i=1}^{n} \ell_{\mathrm{I}}(\mathbf{f}(\mathbf{x}_{\mathrm{I}}^{i}; \mathbf{w}), y_{\mathrm{I}}^{i}) \text{ and } \widehat{R}_{\mathrm{O}}^{sem}(\mathbf{f}) = \frac{1}{m} \sum_{i=1}^{m} \ell_{\mathrm{S}}(\mathbf{f}(\mathbf{x}_{\mathrm{S}}^{i}; \mathbf{w})). \tag{9}$$

Following Hendrycks et al. (2019), we employ the cross entropy loss for $\ell_{\mathrm{I}}$, i.e.,

$$\ell_{\mathrm{I}}(\mathbf{f}(\mathbf{x}_{\mathrm{I}}; \mathbf{w}), y) = -\log \mathtt{sof}_{y} \mathbf{f}(\mathbf{x}_{\mathrm{I}}, \mathbf{w}), \tag{10}$$

and the cross entropy between the uniform distribution and the softmax prediction for distinguishing ID data and the semantic-shifted OOD data, i.e.,

$$\ell_{S}(\mathbf{f}(\mathbf{x}_{\mathrm{S}}; \mathbf{w})) = -\frac{1}{c} \log \mathtt{sof}(\mathbf{f}(\mathbf{x}_{\mathrm{S}}; \mathbf{w})). \tag{11}$$

## 2.3 OPTIMIZATION CONFLICT

Based on the OE manner, we also explore the incorporation of additional covariate-shifted OOD data:

$$S_{\mathrm{O}}^{cov} := \{(\mathbf{x}_{\mathrm{C}}^{1}, y_{\mathrm{I}}), (\mathbf{x}_{\mathrm{C}}^{2}, y_{\mathrm{I}}), \cdots, (\mathbf{x}_{\mathrm{C}}^{t}, y_{\mathrm{I}})\} \overset{\text{i.i.d.}}{\sim} D_{X_{\mathrm{O}}^{cov} Y_{\mathrm{I}}}, \tag{12}$$

during model fine-tuning to endow the model with knowledge about them, thereby enhancing its classification performance on such data. We reformulate the empirical risk after incorporating the covariate-shifted OOD data in below:

$$\begin{aligned} \widehat{R}(\mathbf{f}) &= \widehat{R}_{\mathrm{I}}(\mathbf{f}) + \alpha \cdot \widehat{R}_{\mathrm{O}}^{sem}(\mathbf{f}) + \beta \cdot \widehat{R}_{\mathrm{O}}^{cov}(\mathbf{f}) \\ &= -\frac{1}{n} \sum_{i=1}^{n} [y_{\mathrm{I}} \log(\mathbf{f}(\mathbf{x}_{\mathrm{I}}^{i}; \mathbf{w}))] - \alpha \cdot \frac{1}{m} \sum_{j=1}^{m} \frac{1}{c} \log[(\mathbf{f}(\mathbf{x}_{\mathrm{S}}^{j}; \mathbf{w})] - \beta \cdot \frac{1}{t} \sum_{k=1}^{t} [y_{\mathrm{I}} \log(\mathbf{f}(\mathbf{x}_{\mathrm{C}}^{k}; \mathbf{w}))] \end{aligned}$$
$$\tag{13}$$

Unfortunately, we observe optimizing Eq. (13) directly is difficult due to the existence of a phenomenon called "Optimization Conflict". We formulate such intuition in the following theorem.

**Theorem 1** (Optimization Conflict (OC)). *Given the empirical risk defined in Eq. (13), when optimized via empirical risk minimization, the environment-related components present in both semantically-shifted and covariate-shifted OOD data are driven toward distinct optimization objectives, resulting in an inherent optimization conflict.*

*Proof.* Due to space limitations, we provide the complete proof in Appendix C.1. □

**Remark 3.** *We use $\mathbf{x}_{\mathrm{e}}'$ and $\mathbf{x}_{\mathrm{e}}$ to distinguish the environment-related components in ID data and OOD data, respectively. Furthermore, we consider the OOD data that collected from deployment scenarios, encompassing both semantic-shifted and covariate-shifted OOD data, for model fine-tuning. Thus, the environment-related components in these two datasets are consistent.*

## 3 METHODOLOGY

To address Problem 1 using the OE-based approach while mitigating OC described in Theorem 1, we propose a unified and data-efficient solution in this section with theoretical guarantees. Furthermore, to bridge the gap between theory and practice, we introduce an algorithm named TODG.

## 3.1 Feature Regularization

According to Theorem 1, the source of OC during training lies in the environment-related components of the data. Therefore, a natural approach is to train the model using only the class-related components of the ID data and the covariate-shifted data. Consequently, the key challenge we need to address is how to ensure the model extracts only class-related features while suppressing the extraction of environment-related features. The following theorem provides theoretical guidance for this objective.

**Theorem 2** (Feature Range). *Given an L-layer ReLU network $\mathbf{f}(\cdot; \mathbf{w})$ with the layer parameter $\mathbf{w}^1, \cdots, \mathbf{w}^L$. Assume that the Frobenius norm of the parameter $\mathbf{w}^1, \cdots, \mathbf{w}^L$ are respectively bounded by $\mathcal{A}_1, \cdots, \mathcal{A}_L$, i.e., $\forall l = 1, \cdots, L, \|\mathbf{w}^l\|_F \leq \mathcal{A}_L$. Let $\mathbf{f}(\cdot; \mathbf{w})$ be the feature extractor, and let $D_{X_y}^{fea}$ denotes the feature distribution of data with class label $y$ from the same distribution. Then the variance of the feature distribution $D_{X_y}^{fea}$ is not greater than $\xi_y \prod_{k=1}^K \mathcal{A}_k$, i.e.,*

$$\Sigma \leq \xi_y \prod_{l=1}^L \mathcal{A}_l, \tag{14}$$

*where $\Sigma$ is the variance, and $\xi_y$ is a positive real number.*

*Proof.* We provide the complete proof in Appendix C.2. □

**Remark 4.** *Theorem 2 establishes that the feature representation space of identically distributed data within the same class is upper-bounded, implying these features are confined to a certain domain.*

Based on Theorem 2, we can establish the feature representation model illustrated in Fig. 1, which vividly elucidates the relationship between the features of ID data, covariate-shifted OOD data, and semantic-shifted OOD data. Furthermore, according to this feature representation model, an intuitive approach to mitigating OC is to encourage the model to extract class-related features from covariate-shifted OOD data while suppressing environment-related features. To achieve this, we introduce the following regularization risk term during training.

$$\widehat{R}_{\text{reg}}(\mathbf{f}) = -\sum_{k=1}^t \sum_{j=1}^m \mathtt{KL}(\mathtt{sof}(\mathbf{f}(\mathbf{x}_S^j; \mathbf{w})) \| \mathtt{sof}(\mathbf{f}(\mathbf{x}_C^k; \mathbf{w}))), \tag{15}$$

where $\mathtt{KL}(\cdot\|\cdot)$ represents the KL-divergence. $\widehat{R}_{\text{reg}}(\mathbf{f})$ encourages the model to learn features that maximize the feature distance between covariate-shifted OOD data and semantic-shifted OOD data. According to the feature representation model, this mechanism promotes the extraction of class-related features from covariate-shifted data.

Furthermore, regarding the proposed regularization risk term, we have the following theorem, which ensures that incorporating this risk term can mitigate OC.

**Theorem 3.** *Let $p(\mathbf{x}_{\text{id}})$ denote the probability density function of the class-related feature distribution, and $p(\mathbf{x}_{\text{e}})$ denote that of the environment-related feature distribution. Their joint probability density function is $p(\mathbf{x}_{\text{id}}, \mathbf{x}_{\text{e}})$, with the conditional probability density given by $p(\mathbf{x}_{\text{e}}|\mathbf{x}_{\text{id}})$. Then, the KL-divergence between $p(\mathbf{x}_{\text{e}})$ and $p(\mathbf{x}_{\text{id}}, \mathbf{x}_{\text{e}})$ is proportional to the KL-divergence between $p(\mathbf{x}_{\text{e}})$ and $p(\mathbf{x}_{\text{e}}|\mathbf{x}_{\text{id}})$, namely,*

$$KL(p(\mathbf{x}_{\text{e}}) \| p(\mathbf{x}_{\text{id}}, \mathbf{x}_{\text{e}})) \propto KL(p(\mathbf{x}_{\text{e}}) \| p(\mathbf{x}_{\text{e}}|\mathbf{x}_{\text{id}})). \tag{16}$$

*Proof.* The complete proof is provided in Appendix C.3. □

**Remark 5.** *Theorem 3 establishes the relationship between the $KL(p(\mathbf{x}_{\text{e}}) \| p(\mathbf{x}_{\text{id}}, \mathbf{x}_{\text{e}}))$ and the $KL(p(\mathbf{x}_{\text{e}}) \| p(\mathbf{x}_{\text{e}}|\mathbf{x}_{\text{id}}))$, demonstrating that minimizing $KL(p(\mathbf{x}_{\text{e}}) \| p(\mathbf{x}_{\text{id}}, \mathbf{x}_{\text{e}}))$ is equivalent to minimizing $KL(p(\mathbf{x}_{\text{e}}) \| p(\mathbf{x}_{\text{e}}|\mathbf{x}_{\text{id}}))$. We emphasize that $p(\mathbf{x}_{\text{id}}, \mathbf{x}_{\text{e}})$ is not obtained through direct concatenation of $p(\mathbf{x}_{\text{id}})$ and $p(\mathbf{x}_{\text{e}})$.*

Moreover, based on Theorem 1, the empirical risk term $\widehat{R}_{\mathrm{reg}}(\mathbf{f})$ can be reformulated as:

$$\widehat{R}_{\mathrm{reg}}(\mathbf{f}) = -\sum_{k=1}^{t}\sum_{j=1}^{m}\mathrm{KL}(\mathrm{sof}(\mathbf{f}(\mathbf{x}_{\mathrm{ood}}^{j};\mathbf{w}))\|\mathrm{sof}(\mathbf{f}(\mathbf{x}_{\mathrm{C}}^{k};\mathbf{w}))) \tag{17}$$
$$-\sum_{k=1}^{t}\sum_{j=1}^{m}\mathrm{KL}(\mathrm{sof}(\mathbf{f}(\mathbf{x}_{\mathrm{e}}^{j};\mathbf{w}))\|\mathrm{sof}(\mathbf{f}(\mathbf{x}_{\mathrm{C}}^{k};\mathbf{w}))).$$

Since the class-related components of the semantic-shifted OOD data are unrelated to $\mathrm{KL}(p(\mathbf{x}_{\mathrm{e}})\|p(\mathbf{x}_{\mathrm{e}}|\mathbf{x}_{\mathrm{id}}))$, we treat the first term on the right-hand side of the equation as a constant, $Q$, to analyze the relationship between $\widehat{R}_{\mathrm{reg}}(\mathbf{f})$ and $\mathrm{KL}(p(\mathbf{x}_{\mathrm{e}})\|p(\mathbf{x}_{\mathrm{e}}|\mathbf{x}_{\mathrm{id}}))$. Then, Eq. (17) can be formulated as:

$$\widehat{R}_{\mathrm{reg}}(\mathbf{f}) = -\sum_{k=1}^{t}\sum_{j=1}^{m}\mathrm{KL}(\mathrm{sof}(\mathbf{f}(\mathbf{x}_{\mathrm{e}}^{j};\mathbf{w}))\|\mathrm{sof}(\mathbf{f}(\mathbf{x}_{\mathrm{C}}^{k};\mathbf{w}))) + Q$$
$$= -\sum_{k=1}^{t}\sum_{j=1}^{m}\mathrm{KL}(\mathrm{sof}(\mathbf{f}(\mathbf{x}_{\mathrm{e}}^{j};\mathbf{w}))\|\mathrm{sof}(\mathbf{f}(\mathbf{x}_{\mathrm{id}}^{k};\mathbf{w}))) \tag{18}$$
$$-\sum_{k=1}^{t}\sum_{j=1}^{m}\mathrm{KL}(\mathrm{sof}(\mathbf{f}(\mathbf{x}_{\mathrm{e}}^{j};\mathbf{w}))\|\mathrm{sof}(\mathbf{f}(\mathbf{x}_{\mathrm{e}}^{k};\mathbf{w}))) + Q.$$

According to Theorem 2, $\mathrm{KL}(\mathbf{f}(\mathbf{x}_{\mathrm{e}}^{j};\mathbf{w})\|\mathbf{f}(\mathbf{x}_{\mathrm{e}}^{k};\mathbf{w}))$ is upper-bounded. Therefore, as optimization progresses, we have:

$$\widehat{R}_{\mathrm{reg}}(\mathbf{f}) \propto -\sum_{k=1}^{t}\sum_{j=1}^{m}\mathrm{KL}(\mathrm{sof}(\mathbf{f}(\mathbf{x}_{\mathrm{e}}^{j};\mathbf{w}))\|\mathrm{sof}(\mathbf{f}(\mathbf{x}_{\mathrm{id}}^{k};\mathbf{w}))). \tag{19}$$

Then, with the fact that minimizing $-\mathrm{KL}(p(\mathbf{x}_{\mathrm{e}})\|p(\mathbf{x}_{\mathrm{id}}, \mathbf{x}_{\mathrm{e}}))$ is equivalent to minimizing the right-hand side of the aforementioned proportional relationship. Then, in light of Theorem 3, as optimization progresses, we minimize the environment-related features extracted from the covariate-shifted OOD data, thereby effectively alleviating the OC.

### 3.2 IMPLICIT OOD DATA GENERATION

OE-based methods face another major challenge: requiring additional OOD data for model fine-tuning. This is problematic because deployment scenarios are often unpredictable, making potential OOD samples hard to anticipate, while data collection and annotation are time-consuming and costly. Thus, acquiring sufficient OOD data poses significant practical difficulties.

Fortunately, a small amount of OOD data is available, enabling us to generate sufficient training samples from this small dataset. Based on Theorem 2, features from the same distribution and class exhibit bounded variability. We therefore use an implicit data generation approach to synthesize new OOD samples by sampling from the feature distribution of existing data, rather than creating complete data directly.

Specifically, we utilize the sample mean as an unbiased estimator of the distribution mean and calculate the sample variance based on this mean, i.e., the means can be formalated as:

$$\boldsymbol{\mu}^{cov} = \frac{1}{t}\sum_{k=1}^{t}\mathbf{f}(\mathbf{x}_{\mathrm{C}}^{k};\mathbf{w}), \text{ and } \boldsymbol{\mu}^{sem} = \frac{1}{m}\sum_{j=1}^{m}\mathbf{f}(\mathbf{x}_{\mathrm{S}}^{j};\mathbf{w}). \tag{20}$$

Subsequently, the variances can be formulated as:

$$\Sigma^{cov} = \frac{1}{t}\sum_{k=1}^{t}(\mathbf{f}(\mathbf{x}_{\mathrm{C}}^{k};\mathbf{w}) - \boldsymbol{\mu}^{cov})(\mathbf{f}(\mathbf{x}_{\mathrm{C}}^{k};\mathbf{w}) - \boldsymbol{\mu}^{cov})^{T}, \tag{21}$$

and

$$\Sigma^{sem} = \frac{1}{m}\sum_{j=1}^{m}(\mathbf{f}(\mathbf{x}_{\mathrm{S}}^{j};\mathbf{w}) - \boldsymbol{\mu}^{sem})(\mathbf{f}(\mathbf{x}_{\mathrm{S}}^{j};\mathbf{w}) - \boldsymbol{\mu}^{sem})^{T}. \tag{22}$$

Then, we sample from a Gaussian distribution with the estimated mean and variance, i.e.,

$$\mathbf{v}^{cov} \sim \mathcal{N}(\boldsymbol{\mu}^{cov}, \Sigma^{cov}), \text{ and } \mathbf{v}^{sem} \sim \mathcal{N}(\boldsymbol{\mu}^{sem}, \Sigma^{sem}), \tag{23}$$

where $\mathbf{v}^{cov}$ and $\mathbf{v}^{sem}$ are synthesized covariate- and semantic-shifted OOD features, respectively.

### 3.3 ALGORITHM DETAIL: TODG

We employ the sample mean as an unbiased estimator of the distribution mean. Given the limited availability of two types of OOD data, we use Eq. (20) to estimate the means, $\boldsymbol{\mu}^{cov}$ and $\boldsymbol{\mu}^{sem}$, of the distributions for covariate-shifted OOD data and semantically-shifted OOD data, respectively. Subsequently, we calculate the corresponding variances via Eqs. (21)-(22). We then generate the two types of OOD data via Eq. (23). During the training phase, we train the model by minimizing the empirical risk defined as follows.

$$\widehat{R}(\mathbf{f}) = \widehat{R}_{\mathrm{I}}(\mathbf{f}) + \alpha \cdot \widehat{R}_{\mathrm{O}}^{sem}(\mathbf{f}) + \beta \cdot \widehat{R}_{\mathrm{O}}^{cov}(\mathbf{f}) + \gamma \cdot \widehat{R}_{\mathrm{reg}}(\mathbf{f}). \tag{24}$$

According to Kiryo et al. (2017), a negative loss value tends to induce overfitting. Thus, while preserving the optimization direction, i.e., maximizing the KL-divergence between $\mathtt{sof}(\mathbf{f}(\mathbf{x}_e^j; \mathbf{w}))$ and $\mathtt{sof}(\mathbf{f}(\mathbf{x}_{\mathrm{id}}^k; \mathbf{w}))$, we use the following loss function for $\widehat{R}_{\mathrm{reg}}(\mathbf{f})$ to prevent the occurrence of excessively large negative loss values.

$$\ell_{\mathrm{reg}}(\mathbf{f}(\mathbf{x}_{\mathrm{S}}, \mathbf{x}_{\mathrm{C}}; \mathbf{w})) = -\log \mathtt{KL}(\mathtt{sof}(\mathbf{f}(\mathbf{x}_e^j; \mathbf{w})) \| \mathtt{sof}(\mathbf{f}(\mathbf{x}_{\mathrm{id}}^k; \mathbf{w}))). \tag{25}$$

## 4 EXPERIMENTS

In this section, we evaluate TODG across two tasks. We begin by detailing the evaluation setups.

**Datasets.** We employ ImageNet-200 (Deng et al., 2009) as ID data and ImageNet-200-C (Hendrycks & Dietterich, 2018) as covariate-shifted OOD data. For semantic-shifted OOD data, we utilize real-world natural image datasets: SUN (Xu et al., 2015), iNaturalist (Van Horn et al., 2018), Places365 (Zhou et al., 2018) and Textures (Cimpoi et al., 2014).

**Baselines.** We compare the methods in the fields of OOD detection, including MSP (Hendrycks & Gimpel, 2017), ReAct (Sun et al., 2021), Energy (Liu et al., 2020), MAHA (Lee et al., 2018), MaxLogit (Hendrycks et al., 2022), ODIN (Liang et al., 2018), KNN (Sun et al., 2022), and ASH (Djurisic et al., 2022), and OOD generalization, including ERM (Vapnik, 1999), Mixup (Zhang et al., 2018), IRM (Arjovsky et al., 2019), VREx (Krueger et al., 2021), EQRM (Eastwood et al., 2022), and SharpDRO (Huang et al., 2023). Moreover, we compare with the OE-based methods, which incorporate additional data for fine-tuning, including OE (Hendrycks et al., 2019), Energy(w/OE) (Liu et al., 2020), WOODS (Katz-Samuels et al., 2022), SCONE (Bai et al., 2023), and GTV (Wang & Li, 2024).

**Evaluation Metrics.** We evaluate on the following standard metrics: the classification accuracy of ID data (ID ACC) and covariate-shifted OOD data (OOD ACC); the false positive rate at which semantic-shifted OOD data are declared as ID when 95% of ID and covariate-shifted OOD data are declared as ID (FPR95); and the area under the receiver operating characteristic curve (AUROC).

**TODG Default Setups.** We employ ResNet-18 (He et al., 2016) as the backbone network and fine-tune it using stochastic gradient descent with Nesterov momentum (Duchi et al., 2011). The weight decay coefficient is set to 0.0005, the momentum to 0.09, and the learning rate to 0.001. We fine-tune for 10 epochs with batch sizes of 128 for the ID data, 100 for the covariate-shifted OOD data, and 10 for the semantic-shifted OOD data. Additionally, we set the hyper-parameters to $\alpha = 3$, $\beta = 1$, and $\gamma = 1$. For more training details, please refer to Appendix D.

### 4.1 SYNTHETIC DATASETS

We commence by visualizing both the effects of OC and the performance of our method, TODG, through simulated experiments using multidimensional Gaussian features, where different dimensions represent class-related and environment-related features, the results shown in Fig. 2 indicate that the classical OE method, affected by OC, cannot effectively distinguish data from different distributions. In contrast, TODG with the proposed regularized risk term exhibits better performance. Comprehensive experimental details are presented in Appendix D.

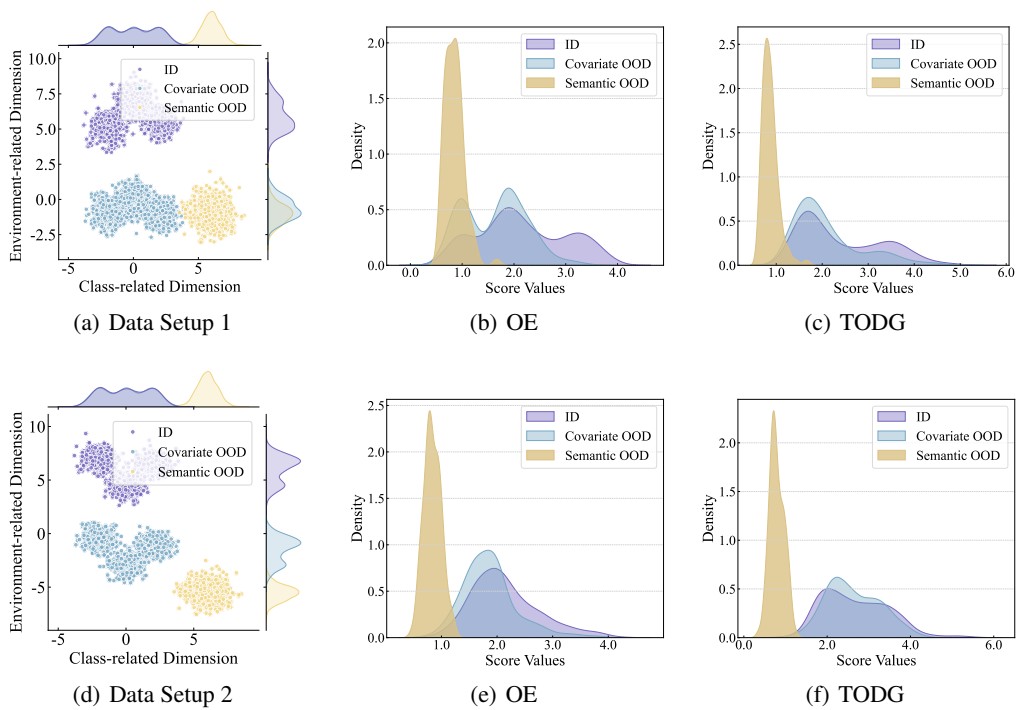

Figure 2: The simulated experiments with multi-dimensional Gaussian features.

| Method | Textures | | Places365 | | iNaturalist | | SUN | | Average Accuracy | |
|---|---|---|---|---|---|---|---|---|---|---|
| | FPR95↓ | AUROC↑ | FPR95↓ | AUROC↑ | FPR95↓ | AUROC↑ | FPR95↓ | AUROC↑ | ID ACC↑ | OOD ACC↑ |
| Methods of OOD Detection | | | | | | | | | | |
| MSP | 44.58 | 88.36 | 70.12 | 79.43 | 42.02 | 86.65 | 53.74 | 85.07 | 85.55 | 62.63 |
| MAHA | 58.16 | 79.25 | 64.31 | 80.16 | 58.53 | 75.03 | 53.52 | 83.11 | 85.55 | 62.63 |
| ODIN | 44.33 | 90.46 | 68.79 | 80.17 | 21.11 | 94.68 | 59.21 | 84.14 | 85.55 | 62.63 |
| Energy | 43.09 | 90.51 | 67.12 | 80.43 | 24.76 | 92.93 | 51.52 | 84.79 | 85.55 | 62.63 |
| ReAct | 42.43 | 91.41 | 57.43 | 82.77 | 59.52 | 79.85 | 64.21 | 79.54 | 85.55 | 62.63 |
| MaxLogit | 43.26 | 90.36 | 53.52 | 81.50 | 38.50 | 86.85 | 57.66 | 81.33 | 85.55 | 62.63 |
| KNN | 24.45 | 95.29 | 64.31 | 81.71 | 24.46 | 93.99 | 52.44 | 83.24 | 85.55 | 62.63 |
| ASH | 24.13 | 95.05 | 61.20 | 84.16 | 21.37 | 95.37 | 57.35 | 84.51 | 85.55 | 62.63 |
| Methods of OOD Generalization | | | | | | | | | | |
| ERM | 40.43 | 87.12 | 67.75 | 78.97 | 41.34 | 85.13 | 55.12 | 85.77 | 85.55 | 62.63 |
| Mixup | 77.43 | 61.34 | 87.96 | 41.21 | 61.52 | 77.21 | 70.44 | 71.53 | 83.14 | 67.57 |
| IRM | 51.78 | 82.44 | 71.52 | 79.41 | 47.24 | 85.10 | 53.15 | 82.33 | 81.01 | 65.41 |
| VREx | 53.92 | 84.73 | 69.45 | 81.17 | 45.12 | 85.57 | 61.92 | 77.82 | 81.24 | 63.17 |
| EQRM | 46.34 | 89.31 | 62.73 | 83.57 | 44.96 | 90.16 | 58.23 | 83.46 | 80.69 | 61.09 |
| SharpDRO | 32.57 | 91.64 | 51.42 | 85.94 | 30.74 | 91.97 | 43.69 | 84.34 | 84.53 | 65.15 |
| Fine-tuning Methods Using Auxiliary Data | | | | | | | | | | |
| OE | 5.91 | 97.23 | 14.74 | 96.44 | 3.74 | 99.12 | 4.17 | 98.72 | 86.17 | 82.61 |
| Energy-OE | 5.26 | 97.78 | 12.72 | 97.02 | 2.58 | 99.41 | 3.65 | 99.13 | 86.95 | 81.44 |
| WOODS | 9.57 | 96.31 | 12.01 | 96.59 | 3.19 | 99.23 | 4.78 | 98.76 | 87.75 | 83.17 |
| SCONE | 10.76 | 96.44 | 13.06 | 96.11 | 5.42 | 98.65 | 4.94 | 98.97 | 87.94 | 83.46 |
| GTV | 7.03 | 97.96 | 11.94 | 96.57 | 2.46 | 99.34 | 3.04 | 99.24 | 88.61 | 83.59 |
| TODG(Ours) | **4.15**$_{\pm 0.8}$ | **98.88**$_{\pm 0.2}$ | **8.90**$_{\pm 0.7}$ | **97.92**$_{\pm 0.3}$ | **1.05**$_{\pm 0.1}$ | **99.58**$_{\pm 0.0}$ | **2.54**$_{\pm 0.0}$ | **99.26**$_{\pm 0.1}$ | **89.19**$_{\pm 0.3}$ | **86.94**$_{\pm 0.5}$ |

Table 1: Comparison between TODG and advanced methods across different semantic-shifted OOD data. ↑ (or ↓) indicates larger or smaller values are preferred. Results are the averages over 10 runs. Bold font indicates the best results in a column.

## 4.2 REAL-WORLD DATASETS

The main results are summarized in Table 1, where `ImageNet-200-C` with snow corruption serves as covariate-shifted OOD data. We highlight the following observations: (a) Methods designed for OOD detection and generalization typically perform well within their respective domains but often exhibit poor performance when applied to other tasks. For instance, methods tailored for OOD

| Method | Gaussian Noise | | Frost | | Fog | | Impulse Noise | |
|---|---|---|---|---|---|---|---|---|
| | ID ACC↑ | OOD ACC↑ | ID ACC↑ | OOD ACC↑ | ID ACC↑ | OOD ACC↑ | ID ACC↑ | OOD ACC↑ |
| OE | 87.34 | 83.43 | 83.26 | 80.15 | 83.60 | 82.62 | 87.52 | 82.29 |
| Energy-OE | 86.23 | 84.92 | 82.70 | 80.67 | 83.24 | 81.26 | 86.35 | 84.65 |
| WOODS | 88.44 | 84.73 | 82.41 | 79.32 | 84.35 | 82.90 | 87.67 | 84.61 |
| SCONE | 88.32 | 84.65 | 83.19 | 80.95 | 85.17 | 82.67 | 88.59 | 84.70 |
| GTV | 89.07 | 85.46 | 83.77 | 81.41 | 84.32 | 83.01 | 87.21 | 84.92 |
| TODG (Ours) | $90.05_{\pm0.2}$ | $87.81_{\pm0.3}$ | $85.15_{\pm0.1}$ | $82.84_{\pm0.1}$ | $86.82_{\pm0.1}$ | $86.25_{\pm0.1}$ | $89.64_{\pm0.4}$ | $88.14_{\pm0.2}$ |

Table 2: Comparison between TODG and advanced methods across covariate-shifted OOD data.

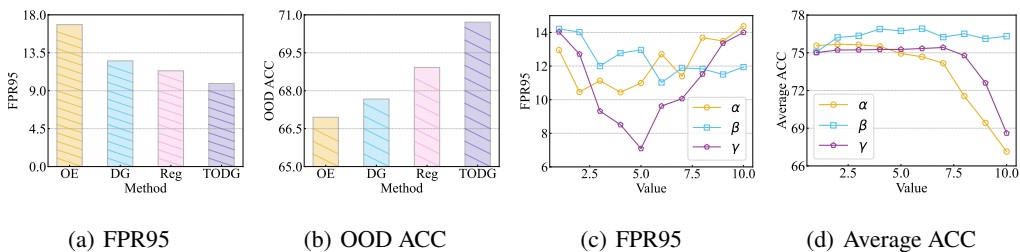

(a) FPR95  (b) OOD ACC  (c) FPR95  (d) Average ACC

Figure 3: In-depth analysis of our proposed method TODG. Figs. 3(a)-3(b) show the results of ablation studies on the two key components, i.e., Data generalization (DG) and Regularization (Reg). Figs. 3(c)-3(d) show the sensitivity analysis of hyper-parameters $\alpha$, $\beta$, and $\gamma$.

generalization generally underperform on OOD detection tasks. (b) Fine-tuning approaches, which leverage additional knowledge, demonstrate superior performance on both tasks, thereby highlighting the advantages of this training strategy. (c) TODG achieves strong OOD detection and generalization performance, outperforming all baselines, including other fine-tuning methods. This is because existing methods fail to address the OC that arise during fine-tuning. In contrast, TODG incorporates a regularization risk term during the optimization process to mitigate these conflicts and employs an implicit data generation strategy to produce more training data. Moreover, as can be seen in Table 2, by resolving OC, TODG further enhances the classification accuracy on covariate-shifted OOD data and consistently outperforms all baselines. Please refer to Appendix E for more results.

## 4.3 IN-DEPTH ANALYSIS

**Ablation Study on Key components.** To perform an in-depth analysis of our proposed TODG and investigate the effectiveness of its individual components, we further conducted ablation studies on the regularization risk term and implicit data generation strategy. The results presented in Figs. 3(a)-3(b) demonstrate that the incorporation of both the regularization risk term and implicitly generated data yields superior performance, confirming their individual effectiveness.

**Hyper-parameter Sensitivity Analysis** As shown in Figs. 3(c)-3(d), our proposed method TODG demonstrates robust performance across wide parameter ranges, eliminating need for meticulous tuning. For example, when $\gamma \in [2, 7]$, increasing $\gamma$ improves both OOD detection performance (FPR95) and classification accuracy on ID and covariate-shifted OOD data (Average ACC).

## 5 CONCLUSION

In this paper, we model the feature representations of ID, covariate-shifted OOD, and semantic-shifted OOD data, revealing the Optimization Conflict (OC) that emerges in OE-based methods when they simultaneously tackle OOD detection and OOD generalization. To mitigate this issue, we propose a unified framework, *Tackling OOD Detection and Generalization* (TODG), which reduces OC through a novel regularization risk term and addresses data scarcity by employing an implicit data generation mechanism during training. Extensive experiments show that TODG consistently outperforms existing baselines, highlighting its effectiveness and reliability. We hope this work offers valuable insights for future research in OOD detection and generalization.

## 6 ETHICS STATEMENT

This study complies with the ICLR Code of Ethics. We propose a novel OOD detection framework and evaluate it on publicly available benchmark datasets. These datasets contain no personally identifiable or sensitive information, thereby ensuring no risks to privacy or security. Our research advances the application of OOD detection in more practical scenarios and holds potential scientific and technological value. All experimental protocols are transparently documented and fairly compared with prior work. The contributions of this study are intended solely for research, supporting the development of artificial intelligence.

## 7 REPRODUCIBILITY STATEMENT

We provide detailed descriptions of our framework, theoretical results, and experimental settings in the paper and appendix. All datasets used are publicly available, and the current description of our method is sufficient for full reproducibility. If the paper is accepted, we will be glad to release the complete implementation to further support the research community.

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

SUPPLEMENTARY MATERIAL: TACKLING OUT-OF-DISTRIBUTION DETECTION AND GENERALIZATION: A UNIFIED AND DATA-EFFICIENT APPROACH

## A  NOTATIONS

In this section, we summarize the key notations in Table 3.

| Notation | Description |
|---|---|
| Variable and Space | |
| $\mathcal{X}$, $\mathcal{Y}$ and $\mathcal{W}$ | feature space, ID label space, and parameter space |
| $X_{\mathrm{I}}$ | ID variable |
| $X_{\mathrm{O}}^{cov}$ and $X_{\mathrm{O}}^{sem}$ | covariate-shifted and semantic-shifted OOD variables |
| $Y_{\mathrm{I}}$ and $Y_{\mathrm{O}}$ | ID and semantic-shifted OOD labels |
| Distribution and Measurement | |
| $D_{X_{\mathrm{I}}Y_{\mathrm{I}}}$ and $D_{X_{\mathrm{O}}^{cov}Y_{\mathrm{I}}}$ | ID joint distribution and covariate-shifted joint distribution |
| $D_{X_{\mathrm{O}}^{sem}}$ | semantic-shifted marginal distribution |
| $D_{XY}$ and $D_X$ | test mixed and marginal distributions |
| $s(\cdot; \mathbf{f})$ and $h(\cdot)$ | OOD scoring function and ID/OOD binary decision function |
| $\mathrm{KL}(\cdot\|\cdot)$ | KL divergence |
| Data and Model | |
| $S_{\mathrm{I}}$, $S_{\mathrm{O}}^{cov}$, and $S_{\mathrm{O}}^{sem}$ | ID training data, covariate-shifted data and semantic-shifted data |
| $n$, $t$, and $m$ | numbers of ID training, covariate-shifted, and semantic-shifted data |
| $\mathbf{x}$, and $y$ | instances and labels |
| $\mathbf{f}(\cdot; \mathbf{w})$ | model outputs |
| Loss and Risk | |
| $\ell_{\mathrm{I}}(\cdot)$ and $\ell_{\mathrm{S}}(\cdot)$ | losses of multi-class classification and ID/OOD binary classification |
| $\ell_{\mathrm{reg}}(\cdot)$ | regularization loss |
| $\widehat{\mathcal{R}}_{\mathrm{I}}(\mathbf{f})$ and $\widehat{\mathcal{R}}_{\mathrm{O}}^{cov}(\mathbf{f})$ | risks of multi-class classification and ID/OOD classification |
| $\widehat{\mathcal{R}}_{\mathrm{A}}^{\mathrm{b}}(\mathbf{f})$ | regularization risk |
| TODG | |
| $\mathbf{x}_{\mathrm{e}}'$, and $\mathbf{x}_{\mathrm{e}}$ | environment-related features |
| $\mathbf{x}_{\mathrm{id}}$, and $\mathbf{x}_{\mathrm{ood}}$ | class-related features |
| $\alpha$, $\beta$, and $\gamma$ | trade-off hyper-parameters |

Table 3: Notations and associated descriptions.

## B  ALGORITHM: TODG

The overall algorithm of our proposed method TODG is presented in Algorithm 1. We employ the sample mean as an unbiased estimator of the distribution mean. Given the limited availability of two types of OOD data, we use Eq. (20) to estimate the means, $\boldsymbol{\mu}^{cov}$ and $\boldsymbol{\mu}^{sem}$, of the distributions for covariate-shifted OOD data and semantically-shifted OOD data, respectively. Subsequently, we calculate the corresponding variances via Eqs. (21)-(22). We then generate the two types of OOD data via Eq. (23). During training, the model is optimized by minimizing the empirical risk in Eq. (24).

## C  PROOFS

In this section, we present complete theoretical proofs for Theorems stated in the main body.

### C.1  PROOF FOR THEOREM 1

To facilitate readability, we restate Theorem 1 here.

**Theorem 1** (Optimization Conflict (OC)). *Given the empirical risk defined in Eq. (13), when optimized via empirical risk minimization, the environment-related components present in both*

---

**Algorithm 1** TODG: **T**ackling **O**OD **D**etection and **G**eneralization

---

**Input**: The ID training dataset $S_{\mathrm{I}}$; the semantic-shifted OOD dataset $S_{X_{\mathrm{O}}^{sem}}$; the convirate-shifted OOD dataset $S_{X_{\mathrm{O}}^{cov}}$; the pre-trained classifier $\mathbf{f}(\cdot; \mathbf{w})$.
**Parameter**: The hyper-parameters $\alpha$, $\beta$ and $\gamma$.
**Output**: The well-trained $\mathbf{f}(\cdot; \mathbf{w})$.

 1: # Implicit OOD Data Generation:
 2: Estimate the distribution means $\boldsymbol{\mu}^{cov}$ and $\boldsymbol{\mu}^{sem}$ via Eq. (20);
 3: Compute the variances $\Sigma^{cov}$ and $\Sigma^{sem}$ via Eqs. (21)-(22);
 4: Generate two types of OOD data via Eq. (23);
 5: # Training:
 6: **for** $epoch \leftarrow 1$ **to** $epochs$ **do**
 7:    Fetch a mini-batch $(\mathcal{B}_{\mathrm{I}}, \mathcal{B}_{\mathrm{O}}^{cov}, \mathcal{B}_{\mathrm{O}}^{sem})$;
 8:    **if** generate OOD data **then**
 9:      Perform steps 2 to 4;
10:    **end if**
11:    Train the model $\mathbf{f}(\cdot; \mathbf{w})$ by minimizing $\widehat{R}(\mathbf{f})$ via Eq. (24);
12: **end for**
13: **return** The well-trained $\mathbf{f}(\cdot; \mathbf{w})$.

---

*semantically-shifted and covariate-shifted OOD data are driven toward distinct optimization objectives, resulting in an inherent optimization conflict.*

*Proof.* To articulate clearly, we explicitly formulate the ID data into two distinct parts, namely,

$$\mathbf{x}_{\mathrm{I}} = \mathbf{x}_{\mathrm{id}} \oplus \mathbf{x}_{\mathrm{e}}', \tag{26}$$

where the symbol $'\oplus'$ indicates that the two terms on the right-hand side collectively constitute the left-hand side. $\mathbf{x}_{\mathrm{I}}$ and $\mathbf{x}_{\mathrm{e}}'$ represent the class-related and enviroment-related components, respectively. Similarly, the semantic-shifted and coviriate-shifted OOD data can also be formulated as follows.

$$\mathbf{x}_{\mathrm{S}} = \mathbf{x}_{\mathrm{ood}} \oplus \mathbf{x}_{\mathrm{e}}, \text{ and } \mathbf{x}_{\mathrm{C}} = \mathbf{x}_{\mathrm{id}} \oplus \mathbf{x}_{\mathrm{e}}. \tag{27}$$

Subsequently, the empirical risk can be reformulated as:

$$\begin{aligned}
\widehat{R}(\mathbf{f}) &= \widehat{R}_{\mathrm{I}}(\mathbf{f}) + \alpha \cdot \widehat{R}_{\mathrm{O}}^{sem}(\mathbf{f}) + \beta \cdot \widehat{R}_{\mathrm{O}}^{cov}(\mathbf{f}) \\
&= -\frac{1}{n} \sum_{i=1}^{n} [y_{\mathrm{I}} \log \mathtt{sof}(\mathbf{f}(\mathbf{x}_{\mathrm{id}}^i; \mathbf{w})) + y_{\mathrm{I}} \log \mathtt{sof}(\mathbf{f}(\mathbf{x}_{\mathrm{e}}'^i; \mathbf{w}))] \\
&\quad -\frac{\beta}{t} \sum_{k=1}^{t} [y_{\mathrm{I}} \log \mathtt{sof}(\mathbf{f}(\mathbf{x}_{\mathrm{id}}^k; \mathbf{w})) + \underbrace{y_{\mathrm{I}} \log \mathtt{sof}(\mathbf{f}(\mathbf{x}_{\mathrm{e}}^k; \mathbf{w}))}_{(1)}] \\
&\quad -\frac{\alpha}{m} \sum_{j=1}^{m} [\frac{1}{c} \log \mathtt{sof}(\mathbf{f}(\mathbf{x}_{\mathrm{ood}}^j; \mathbf{w})) + \underbrace{\frac{1}{c} \log \mathtt{sof}(\mathbf{f}(\mathbf{x}_{\mathrm{e}}^j; \mathbf{w}))}_{(2)}],
\end{aligned} \tag{28}$$

where $\mathbf{x}_{\mathrm{e}}$ is optimized toward different objectives in $(1)$ and $(2)$, leading to an optimization conflict. Thus, we complete the proof. $\qquad\square$

### C.2 PROOF FOR THEOREM 2

We also restate Theorem 2 here.

**Theorem 2** (Feature Range). *Given an $L$-layer ReLU network $\mathbf{f}(\cdot; \mathbf{w})$ with the layer parameter $\mathbf{w}^1, \cdots, \mathbf{w}^L$. Assume that the Frobenius norm of the parameter $\mathbf{w}^1, \cdots, \mathbf{w}^L$ are respectively bounded by $\mathcal{A}_1, \cdots, \mathcal{A}_L$, i.e., $\forall l = 1, \cdots, L, \|\mathbf{w}^l\|_F \leq \mathcal{A}_L$. Let $\mathbf{f}(\cdot; \mathbf{w})$ be the feature extractor, and let $D_{X_y}^{fea}$ denotes the feature distribution of data with class label $y$ from the same distribution. Then the variance of the feature distribution $D_{X_y}^{fea}$ is not greater than $\xi_y \prod_{k=1}^{K} \mathcal{A}_k$, i.e.,*

$$\Sigma \leq \xi_y \prod_{l=1}^{L} \mathcal{A}_l, \tag{29}$$

*where $\Sigma$ is the variance, and $\xi_y$ is a positive real number.*

**Assumption 1.** *For any two samples drawn from the same distribution and belonging to the same class, the distance between them is not greater than $\xi_y$, i.e.,*

$$max\|\mathbf{x}^i - \mathbf{x}^j\|_F \le \xi_y, \ \mathbf{x}^i, \mathbf{x}^j \overset{i.i.d.}{\sim} D_{XY=y}, \tag{30}$$

*where $\|\cdot\|_F$ is the Frobenius norm.*

**Remark 6.** *This assumption serves as a foundational premise in machine learning. It is the validity of this assumption that enables tasks such as image recognition, classification, and related applications. Therefore, a significant body of research is built upon this underlying assumption (Wang et al., 2025; Gong et al., 2021).*

Under Assumption 1, we prove Theorem 2.

*Proof.* Consider an $L$-layer network formulated as follows.

$$\mathbf{f}(\mathbf{x}; \mathbf{w}) = \mathbf{w}^l \sigma \circ (\mathbf{w}^{l-1} \sigma \circ (\cdots \sigma \circ (\mathbf{w}^1 \mathbf{x}))),$$

where we assume the activation function $\sigma$ is 1-Lipschitz (holding for ReLU). It is worth noting that convolution operations can also be represented through matrix multiplication. We utilize the sample mean as an unbiased estimator of the distribution mean.

When $l = 1$, the mean of feature distribution $D_{X_y}^{fea}$ is :

$$\boldsymbol{\mu} = \frac{1}{n} \sum_{i=1}^{n} \sigma \circ \mathbf{w}_1 \cdot \mathbf{x}^i,$$

where $n$ is the number the samples. Then, the variance of the feature distribution $D_{X_y}^{fea}$ can be computed as follows.

$$\Sigma = \frac{1}{n} \sum_{i=1}^{n} \|\sigma \circ \mathbf{w}^1 \cdot \mathbf{x}^i - \boldsymbol{\mu}\|_F$$

$$= \frac{1}{n} \sum_{i=1}^{n} \|\sigma \circ \mathbf{w}^1 \cdot \mathbf{x}^i - \frac{1}{n} \sum_{j=1}^{n} \sigma \circ \mathbf{w}^1 \cdot \mathbf{x}^j\|_F$$

$$= \frac{1}{n} \sum_{i=1}^{n} \|\frac{1}{n} \sum_{j \ne i} (\sigma \circ \mathbf{w}^1 \cdot \mathbf{x}^j - \sigma \circ \mathbf{w}^1 \cdot \mathbf{x}^i)\|_F$$

$$\le \frac{1}{n^2} \sum_{i=1}^{n} \sum_{j \ne i} \|(\sigma \circ \mathbf{w}^1 \cdot \mathbf{x}^j - \sigma \circ \mathbf{w}^1 \cdot \mathbf{x}^i)\|_F$$

$$\le \frac{1}{n^2} \sum_{i=1}^{n} \sum_{j \ne i} \|(\mathbf{w}^1 \cdot \mathbf{x}^i - \mathbf{w}^1 \cdot \mathbf{x}^j)\|_F$$

$$= \frac{1}{n^2} \sum_{i=1}^{n} \sum_{j \ne i} \|\mathbf{w}^1 \cdot (\mathbf{x}^i - \mathbf{x}^j)\|_F$$

$$\le \frac{1}{n^2} \sum_{i=1}^{n} \sum_{j \ne i} \|\mathbf{w}^1\|_F \cdot \|\mathbf{x}^i - \mathbf{x}^j\|_F$$

$$\le \xi_y \mathcal{A}_1$$

It is obvious that when $l = 2$, there is

$$\Sigma \le \xi_c \|\mathbf{w}^1\|_F \|\mathbf{w}^2\|_F \le \xi_y \mathcal{A}_1 \mathcal{A}_2.$$

According to mathematical induction, when $l = L$, there is

$$\Sigma \le \xi_y \prod_{l=1}^{L} \|\mathbf{w}^l\|_F \le \xi_y \prod_{l=1}^{L} \mathcal{A}_l.$$

Thus, the proof is completed. □

### C.3 PROOF FOR THEOREM 3

We also restate Theorem 3 here.

**Theorem 3.** *Let $p(\mathbf{x}_{id})$ denote the probability density function of the class-related feature distribution, and $p(\mathbf{x}_e)$ denote that of the environment-related feature distribution. Their joint probability density function is $p(\mathbf{x}_{id}, \mathbf{x}_e)$, with the conditional probability density given by $p(\mathbf{x}_e|\mathbf{x}_{id})$. Then, the KL-divergence between $p(\mathbf{x}_e)$ and $p(\mathbf{x}_{id}, \mathbf{x}_e)$ is proportional to the KL-divergence between $p(\mathbf{x}_e)$ and $p(\mathbf{x}_e|\mathbf{x}_{id})$, namely,*

$$KL(p(\mathbf{x}_e)\|p(\mathbf{x}_{id}, \mathbf{x}_e)) \propto KL(p(\mathbf{x}_e)\|p(\mathbf{x}_e|\mathbf{x}_{id})). \tag{31}$$

*Proof.* According to the Bayes' theorem, we have

$$p(\mathbf{x}_{id}) = \frac{p(\mathbf{x}_{id}, \mathbf{x}_e)}{p(\mathbf{x}_e|\mathbf{x}_{id})}.$$

Taking the logarithm of both sides of the equation, we obtain:

$$\log p(\mathbf{x}_{id}) = \log p(\mathbf{x}_{id}, \mathbf{x}_e) - \log p(\mathbf{x}_e|\mathbf{x}_{id})$$
$$= [\log p(\mathbf{x}_{id}, \mathbf{x}_e) - \log p(\mathbf{x}_e)] - [\log p(\mathbf{x}_e|\mathbf{x}_{id}) - \log p(\mathbf{x}_e)]$$
$$= \log \frac{p(\mathbf{x}_{id}, \mathbf{x}_e)}{p(\mathbf{x}_e)} - \log \frac{p(\mathbf{x}_e|\mathbf{x}_{id})}{p(\mathbf{x}_e)}.$$

Multiplying both sides of the equation by $p(\mathbf{x}_e)$ and integrating with respect to $\mathbf{x}_e$. For the left-hand side of the equation, we have:

$$\int_{\mathbf{x}_e} p(\mathbf{x}_e) \log p(\mathbf{x}_{id}) d\mathbf{x}_e = \log p(\mathbf{x}_{id}) \int_{\mathbf{x}_e} p(\mathbf{x}_e) d\mathbf{x}_e = \log p(\mathbf{x}_{id}).$$

And for the right-hand side of the equation, we have:

$$\int_{\mathbf{x}_e} p(\mathbf{x}_e) \log \frac{p(\mathbf{x}_{id}, \mathbf{x}_e)}{p(\mathbf{x}_e)} d\mathbf{x}_e - \int_{\mathbf{x}_e} p(\mathbf{x}_e) \log \frac{p(\mathbf{x}_e|\mathbf{x}_{id})}{p(\mathbf{x}_e)} d\mathbf{x}_e$$
$$= -KL(p(\mathbf{x}_e)\|p(\mathbf{x}_{id}, \mathbf{x}_e)) + KL(p(\mathbf{x}_e)\|p(\mathbf{x}_e|\mathbf{x}_{id})).$$

Under fixed samples, $\log p(\mathbf{x}_{id})$ is a constant term. This implies an proportionality between $KL(p(\mathbf{x}_e)\|p(\mathbf{x}_{id}, \mathbf{x}_e))$ and $KL(p(\mathbf{x}_e)\|p(\mathbf{x}_e|\mathbf{x}_{id}))$ when conditioned on available samples, i.e.,

$$KL(p(\mathbf{x}_e)\|p(\mathbf{x}_{id}, \mathbf{x}_e)) \propto KL(p(\mathbf{x}_e)\|p(\mathbf{x}_e|\mathbf{x}_{id})).$$

Thus, the proof is completed. □

## D  ADDITIONAL TRAINING DETAILS

**Pre-training Setups.** For the experiments on `CIFAR` benchmarks, we utilize Wide-ResNet-40-2 (Zagoruyko & Komodakis, 2016) as the backbone network. It is trained for 200 epochs using empirical risk minimization, with a batch size of 64, momentum of 0.9, and an initial learning rate of 0.1. The learning rate is divided by a factor of 10 at epochs 100 and 150. For the experiments on `ImageNet-200`, we employ ResNet-18 (He et al., 2016) as the backbone network. The pre-training follows the schedule in Yang et al. (2022), where the model is trained for 100 epochs with cross-entropy loss. The optimizer is SGD with a momentum of 0.9, an initial learning rate of 0.1, and a cosine annealing decay schedule (Loshchilov & Hutter, 2017). Additionally, a weight decay of 0.0005 is applied.

| Semantic-Shifted Data | Method | FPR95 ↓ | AUROC ↑ | ID ACC. ↑ | OOD ACC. ↑ |
|---|---|---|---|---|---|
| Textures | OE | 71.62 | 80.34 | 86.17 | 82.61 |
| | Energy(w/OE) | 69.82 | 81.90 | 86.91 | 81.44 |
| | WOODS | 67.78 | 80.76 | 87.75 | 83.17 |
| | SCONE | 69.23 | 79.25 | 87.94 | 83.46 |
| | GTV | 67.52 | 81.60 | 88.61 | 83.59 |
| | TODG (Ours) | 65.82 | 82.24 | 89.19 | 86.94 |
| iNaturalist | OE | 20.77 | 94.21 | 86.17 | 82.61 |
| | Energy(w/OE) | 20.56 | 94.77 | 86.91 | 81.44 |
| | WOODS | 20.92 | 94.96 | 87.75 | 83.17 |
| | SCONE | 21.81 | 94.14 | 87.94 | 83.46 |
| | GTV | 19.26 | 94.33 | 88.61 | 83.59 |
| | TODG (Ours) | 18.50 | 95.19 | 89.19 | 86.94 |
| SUN | OE | 10.12 | 96.35 | 86.17 | 82.61 |
| | Energy(w/OE) | 9.71 | 97.13 | 86.91 | 81.44 |
| | WOODS | 8.65 | 97.63 | 87.75 | 83.17 |
| | SCONE | 10.21 | 97.20 | 87.94 | 83.46 |
| | GTV | 9.93 | 97.96 | 88.61 | 83.59 |
| | TODG (Ours) | 7.49 | 98.21 | 89.19 | 86.94 |

Table 4: Comparison between TODG and advanced methods across different unseen semantic-shifted OOD data. ↑ (or ↓) indicates larger or smaller values are preferred.

**OOD Scoring.** We use MLP scores as the OOD score during the testing phase. Moreover, the results presented in Table 7 demonstrate that TODG achieves satisfactory performance when utilizing a variety of distinct scores.

**Number of Data.** For `CIFAR` benchmarks, we use 500 images for the semantic-shifted OOD data. For the covariate-OOD data, We utilized one-tenth of the data volume per class in the ID dataset.

**Data Generation.** In each batch iteration, for the covariate-shifted OOD data, we generate five samples for each class present in that batch. Meanwhile, for the semantic-shifted OOD data, we generate 100 samples.

**Details of the Simulated Experiments.** We use different dimensions of a two-dimensional Gaussian distribution to represent the class-related and environment-related features, respectively. Specifically, the covariate-shifted OOD data is identical to ID data in terms of the class-related dimension, yet diverges in the environment-related dimension. We consider two different data setups as follows.

- **Date setup 1:** As shown in Fig. 4(a) in the main body, the ID data is generated from three distinct multivariate Gaussian distributions, each corresponding to one of the three classes, with mean vectors of $[-2.0, 5.0], [0, 7.0]$ and $[2.0, 5.5]$. The mean vectors of covariate-shifted OOD data are $[-2.0, -1.0], [0, 0]$ and $[2.0, -1.0]$. Moreover, we generate the semantic-shifted OOD data from the multivariate Gaussian distribution $\mathcal{N}([6.0, -1.0], 0.55 \cdot \mathbf{I})$ where $\mathbf{I}$ is a $2 \times 2$ identity matrix.

- **Data setup 2:** As shown in Fig. 4(d) in the main body, the ID data is generated from three distinct multivariate Gaussian distributions, each corresponding to one of the three classes, with mean vectors of $[-2.0, 7.0], [0, 4.5]$ and $[2.0, 6.5]$. The mean vectors of covariate-shifted OOD data are $[-2.0, -0.5], [0, -3.0]$ and $[2.0, -1.0]$. Moreover, we generate semantic-shifted OOD data from the multivariate Gaussian distribution $\mathcal{N}([6.0, -5.5], 0.55 \cdot \mathbf{I})$ where $\mathbf{I}$ is a $2 \times 2$ identity matrix.

In both setups, all classes of ID and covariate-shifted OOD data share the same covariance matrix $\begin{bmatrix} 0.35 & 0 \\ 0 & 0.35 \end{bmatrix}$. We generate 450 samples for each class. We conduct this experiment by using a three-layer fully connected neural network.

**Additional details for the Ablation Study.** To perform an in-depth analysis of TODG and investigate the effectiveness of its individual components, we further conducted ablation studies on the regulariza-

| Covariate Shift Type | Method | FPR95 ↓ | AUROC ↑ | ID ACC. ↑ | OOD ACC. ↑ |
|---|---|---|---|---|---|
| Gaussian Noise | GTV | 9.56 | 97.90 | 89.07 | 85.46 |
| | TODG (Ours) | 7.54 | 98.06 | 90.05 | 87.81 |
| Defocus Blur | GTV | 12.59 | 96.78 | 86.41 | 84.32 |
| | TODG (Ours) | 10.37 | 97.86 | 88.76 | 86.20 |
| Fog | GTV | 9.62 | 97.23 | 84.32 | 83.01 |
| | TODG (Ours) | 9.26 | 97.93 | 86.82 | 86.25 |
| Frost | GTV | 13.44 | 96.12 | 83.77 | 81.41 |
| | TODG (Ours) | 11.02 | 97.71 | 85.15 | 82.84 |
| Gaussian Blur | GTV | 11.46 | 97.23 | 88.10 | 85.31 |
| | TODG (Ours) | 11.27 | 97.42 | 89.70 | 88.99 |
| Glass Blur | GTV | 10.77 | 95.40 | 87.23 | 84.14 |
| | TODG (Ours) | 7.60 | 97.96 | 88.15 | 88.28 |
| Impulse Noise | GTV | 8.73 | 97.65 | 89.21 | 84.92 |
| | TODG (Ours) | 7.37 | 98.13 | 89.64 | 88.14 |
| Jpeg Compression | GTV | 8.44 | 97.31 | 89.95 | 85.60 |
| | TODG (Ours) | 8.36 | 97.76 | 89.98 | 89.08 |
| Motion Blur | GTV | 14.43 | 95.67 | 87.92 | 83.55 |
| | TODG (Ours) | 11.57 | 97.51 | 89.38 | 88.56 |
| Saturate | GTV | 26.31 | 93.62 | 83.91 | 75.64 |
| | TODG (Ours) | 22.23 | 95.94 | 91.00 | 80.05 |
| Shot Noise | GTV | 8.01 | 97.60 | 87.76 | 85.13 |
| | TODG (Ours) | 7.92 | 98.08 | 88.32 | 87.38 |
| Spatter | GTV | 11.52 | 96.21 | 88.72 | 85.69 |
| | TODG (Ours) | 10.51 | 97.61 | 90.23 | 90.39 |
| Speckle Noise | GTV | 9.60 | 97.77 | 84.31 | 81.60 |
| | TODG (Ours) | 7.38 | 98.06 | 88.72 | 85.94 |
| Zoom Blur | GTV | 12.79 | 95.10 | 85.93 | 84.23 |
| | TODG (Ours) | 9.93 | 97.73 | 88.30 | 87.29 |
| Contrast | GTV | 8.71 | 97.25 | 85.03 | 83.77 |
| | TODG (Ours) | 7.78 | 98.05 | 89.42 | 86.34 |

Table 5: Comparison between TODG and SOTA method across different covariate-shifted OOD data.

tion risk term and implicit data generation strategy. We employ CIFAR-100 (Krizhevsky & Hinton, 2009) as the ID dataset, CIFAR-100-C (Hendrycks & Dietterich, 2018) as the covariate-shifted OOD dataset, and Textures as the semantic-shifted OOD dataset.

**Additional Details for Parameter Sensitivity.** We used CIFAR-100 as ID data, CIFAR-100-C as covariate-shifted OOD data, and Places365 as semantic-shifted OOD data. The metric Average ACC is the average classification accuracy on ID and covariate-shifted OOD data.

**Details for the Experiments of Table 2.** To further investigate the performance of TODG, extensive experiments are conducted on datasets with different covariate shifts, where the Places365 serves as the semantic-shifted OOD data.

## E  ADDITIONAL RESULTS AND ANALYSIS

**Unseen Semantic-shifted OOD Data.** To further investigate the performance of our method, TODG, we conduct experiments on more challenging unseen semantic-shifted data. Specifically, we

| Method | FPR95 ↓ | AUROC ↑ | ID ACC ↑ | OOD ACC ↑ |
|--------|---------|---------|----------|-----------|
| OE | 34.51 | 88.75 | 80.72 | 64.91 |
| Energy | 33.46 | 87.26 | 81.77 | 64.78 |
| WOODS | 32.53 | 88.73 | 81.64 | 67.52 |
| SCONE | 32.01 | 87.62 | 81.50 | 67.57 |
| GTV | 31.42 | 89.16 | 82.17 | 67.65 |
| TODG | **29.86** | **89.81** | **82.29** | **69.31** |

Table 6: Comparison between TODG and advanced SOTA methods on textual data.

use `ImageNet-200` as the ID data, `ImageNet-200-C` as the covariate-shifted OOD data, and `Places365` as the semantic-shifted OOD data. Additionally, we test on three other semantic-shifted OOD datasets. The experimental results are presented in Table 4. As shown, our method consistently outperforms other methods, demonstrating strong reliability and robustness.

**Additional Results on Textual Data.** we conduct additional experiments on textual data using the classical `20 Newsgroups` dataset. In textual data, class-related components are words reflecting the target category, while environment-related components are contextual words. We used 15 categories from `20 Newsgroups` as ID data and the remaining five as semantic-shift OOD data. Covariate-shift OOD data was generated from ID data via different processing methods. For both OOD types, only 10% of samples were used. The results are presented in Table 6, as can be seen, our method remains effective on textual data and outperforms all baseline methods. These results demonstrate that our method is effective not only for image data but also for textual data.

**OOD Scoring.** We evaluate the OOD detection performance of a well-trained model with TODG using various OOD scores. Specifically, the ID data is `ImageNet-200`, the covariate-shifted OOD data is `ImageNet-200-C`, and the semantic-shifted OOD data is `Places365`. The experimental results are summarized in Table 7. As we can see, the model trained with TODG demonstrates robust OOD detection performance across multiple OOD scores.

| Method | FPR95 ↓ | AUROC ↑ |
|--------|---------|---------|
| MSP | 9.01 | 98.03 |
| Energy | 8.18 | 97.81 |
| MLP | 8.90 | 97.92 |

Table 7: Evaluating the OOD detection performance of TODG using various OOD scores.

**Various Covariate-shifted OOD data.** To provide a more thorough assessment of OOD generalization performance of our TODG, we evaluate its performance across 16 diverse covariate-shifted OOD datasets. We maintain the use of `ImageNet-200` as the ID data and utilize `Places365` as the semantic-shifted OOD data. The experimental outcomes are detailed in Table 5. TODG consistently exhibits superior performance in OOD generalization tasks compared to the latest SOTA method, GTV, thereby further demonstrating its superior performance.

**Feature Visualization.** Extensive experiments have demonstrated the robust performance of our proposed method, TODG, in OOD detection and generalization tasks, thereby corroborating our theoretical analysis which indicates that the regularization risk term we proposed enables the model to extract class-related features from covariate-shifted data while ignoring environment-related features during training. To further investigate this, we employ t-SNE to visualize the features extracted by the model for ID, covariate-shifted OOD, and semantic-shifted OOD data after training completion, using the outputs of penultimate layer of the model as the extracted features. We utilize `CIFAR-10` as ID data, `CIFAR-10-C` with snow corruption as the covariate-shifted OOD data, and `Places365` as the semantic-shifted OOD data. The visualization results are presented in Fig. 4. Compared with models trained using the classic OE, the features extracted by models trained with TODG show more compact overlap between ID and covariate-shifted OOD data, thereby validating the effectiveness of our proposed regularization risk term.

**Additional Results on Adversarial Generated OOD Data.** We conduct additional experiments on adversarially generated OOD data. We generate adversarial examples for `CIFAR-100` using `PGD` (Madry et al., 2018), targeting the model `Wide-ResNet-40-2`, and use them as covariate-

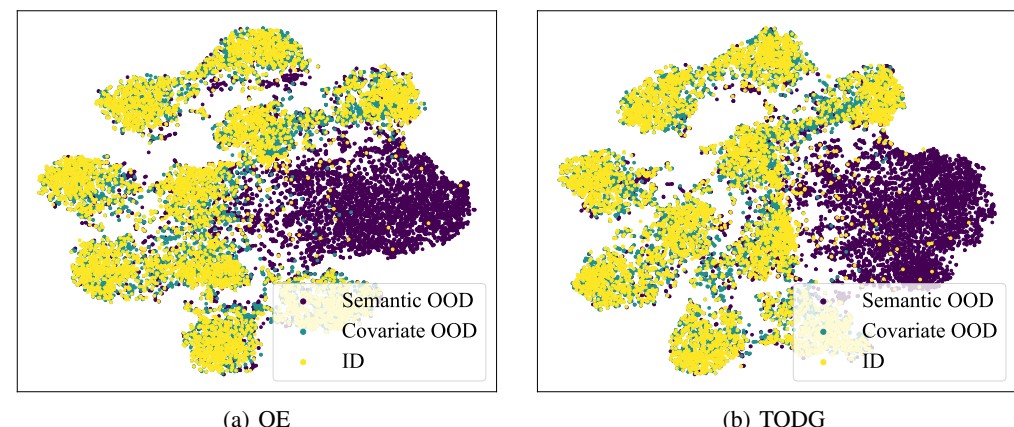

|          | (a) OE          |              | (b) TODG     |
|----------|-----------------|--------------|--------------|

Figure 4: The T-SEN feature visualization results on the `CIFAR-10` datasets. Fig. 4(a) shows the feature visualization without the regularization risk term and implicit data generation. In contrast, Fig. 4(b) displays the result of TODG, which incorporates these two strategies.

| Method | FPR95↓ | AUROC↑ | ID ACC↑ | OOD ACC↑ |
|--------|--------|--------|---------|----------|
| OE     | 37.63  | 92.75  | 70.60   | 50.79    |
| Energy | 36.41  | 91.60  | 71.92   | 50.82    |
| WOODS  | 32.90  | 92.78  | 72.16   | 51.34    |
| SCONE  | 34.46  | 91.42  | 71.56   | 51.69    |
| GTV    | 31.79  | 92.65  | 72.26   | 52.37    |
| TODG   | **27.54** | **94.83** | **72.44** | **53.02** |

Table 8: Comparison between TODG and advanced methods on adversarial generated OOD data.

shift OOD data, with `Places365` as semantic-shift OOD data. The results in Tabel 8 show that TODG remains effective on adversarially generated OOD data and outperforms all baseline methods, further demonstrating its robustness.

**Varying Quantities of OOD Data.** We conduct additional experiments to investigate the impact of varying quantities of OOD data. We use `CIFAR-100` as the ID data, `CIFAR-100-C` with snow as the corrupt type as covariate-shift OOD data, and `Places365` as semantic-shift OOD data. The results are presented in Tabel 9. As the quantity of OOD samples increases, the performance of our method improves. Futhermore, the results also demonstrate that TODG remains effective even with limited OOD samples.

**Additional Analysis of Ablation Study.** We conduct a comprehensive analysis of the simulated experiments presented in the main body of this paper. In these experiments, data setup 1 represents the scenario where the environment-related features of covariate-shifted and semantic-shifted OOD data are nearly overlapping, while data setup 2 represents the scenario with minimal overlap. It is worth noting that, since we aim to assign higher OOD scores to ID data and lower OOD scores to semantic-shifted OOD data, the distribution of OOD scores for semantic-shifted OOD data should have minimal overlap with the distributions corresponding to ID and covariate-shifted OOD data. We highlight the following observations: (a) When the overlap of environment-related features is minimal, both the classic OE and TODG methods exhibit strong performance. (b) When the overlap of environment-related features is high, the classic OE method results in significant overlap among the OOD scores of the three distributions, leading to poor OOD detection performance and weak discrimination ability for ID and covariate-shifted OOD data. In contrast, TODG not only demonstrates superior OOD detection capabilities but also achieves nearly overlapping score distributions for ID and covariate-shifted OOD data. This indicates that our method enhances the model ability to extract class-related features from covariate-shifted OOD data, thereby alleviating the optimization conflict. Therefore, the experimental results support our theoretical analysis.

| Number | FPR95 ↓ | AUROC ↑ | ID ACC ↑ | OOD ACC ↑ |
|--------|---------|---------|----------|-----------|
| 10 | 18.22 | 96.02 | 76.01 | 65.72 |
| 20 | 12.85 | 97.19 | 76.23 | 67.44 |
| 30 | 9.58 | 97.88 | 76.41 | 70.76 |
| 40 | 9.04 | 97.42 | 76.65 | 72.20 |
| 50 | 8.96 | 97.95 | 78.19 | 72.42 |

Table 9: Experiments on varying quantities of OOD data.

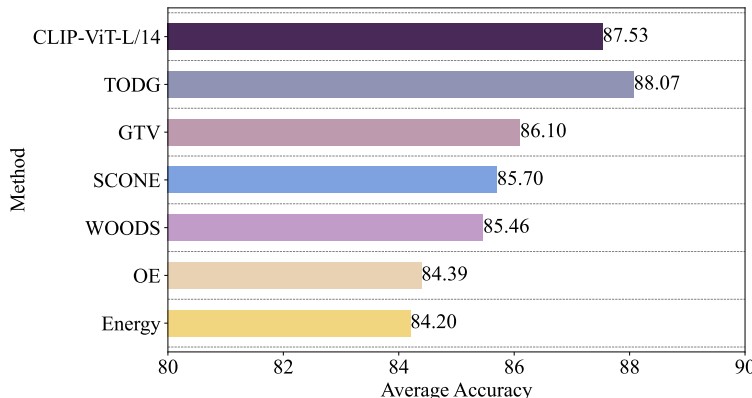

Figure 5: The experiments compare the OOD Generalization performance to CLIP-ViT. The results are the average accuracy on ID and covariate-shifted OOD data.

**Compare to CLIP-VIT.** Vision-language models, such as CLIP (Radford et al., 2021), which are trained on extensive image-text pairs, have notably exhibited strong generalization capabilities against various distribution shifts (Addepalli et al., 2024). We are interested in comparing the generalization abilities of ResNet-18, trained solely on image data, and the visual encoder within CLIP. To this end, we compare the OOD generalization performance of ResNet-18 trained with different methods and CLIP-ViT-L/14. We utilize `ImageNet-200` as the ID data, `ImageNet-200-C` with snow corruption as the covariate-shifted OOD data, and `Places365` as the semantic-shifted OOD data. The experimental results are shown in Fig. 5. As we can see, TODG achieves superior OOD generalization performance. Although the experiment is not perfectly rigorous, it still reflects the strong performance of our approach.

# F  DATASETS

In this section, we present a detailed description of the datasets employed in this paper.

**CIFAR-10** (Krizhevsky & Hinton, 2009) is a widely-used dataset, consisting of 60,000 32×32 color images across 10 distinct classes. It is divided into a training set with 50,000 images and a test set with 10,000 images. The experimental results on this dataset is provided in Section E.

**CIFAR-100** (Krizhevsky & Hinton, 2009) is a significant extension of the `CIFAR-10` dataset, consisting of 60,000 32×32 color images divided into 100 classes, with 600 images per class. Each class contains 500 training images and 100 testing images. The experimental results on this dataset is provided in Section E.

**CIFAR-10-C** (Hendrycks & Dietterich, 2018) is specifically designed to evaluate model robustness, derived from `CIFAR-10` by applying various corruptions, including gaussian noise, defocus blur, glass blur, impulse noise, shot noise, snow, and zoom blur.

**CIFAR-100-C** (Hendrycks & Dietterich, 2018) is analogous to `CIFAR-10-C`, but it is derived from `CIFAR-100`.

**ImageNet-200** (Deng et al., 2009) is the subset of ImageNet-1k, containing 200 distinct classes. We present the class labels along with their corresponding WordNet IDs as follows.

n04133789 n01531178 n01498041 n03498962 n02088094 n07697313 n04591713 n07718472
n07714990 n04275548 n04141076 n01518878 n01860187 n02233338 n03345487 n02950826
n02110958 n02325366 n01630670 n02483362 n02808440 n02655020 n07695742 n01806143
n04465501 n02797295 n02051845 n01855672 n04086273 n07697537 n02088466 n02119022
n02966193 n04192698 n01770393 n04389033 n03676483 n03888257 n02510455 n02123045
n02363005 n03710193 n07753592 n01944390 n03947888 n01484850 n07693725 n02883205
n02108089 n01774750 n02096585 n01843383 n02112018 n02114367 n02106662 n04325704
n02110341 n02056570 n07714571 n02117135 n02099712 n01494475 n04409515 n01616318
n07749582 n02071294 n07880968 n02098286 n02749479 n02279972 n02814860 n02206856
n02128757 n03775071 n02793495 n03272010 n02391049 n02843684 n02346627 n02097298
n10565667 n03649909 n04266014 n04146614 n01882714 n07753275 n07768694 n02190166
n02085620 n02948072 n04310018 n03124170 n02108915 n01986214 n02398521 n02088364
n04487394 n02106550 n03602883 n01820546 n02130308 n03424325 n02007558 n04254680
n02447366 n02395406 n02138441 n02802426 n02066245 n02009912 n07873807 n01694178
n02992529 n02102318 n02165456 n03594945 n02134084 n02086240 n07734744 n02769748
n02236044 n01983481 n07614500 n07745940 n03452741 n02909870 n02099601 n03630383
n02364673 n02526121 n02268443 n02480855 n02980441 n02129604 n03372029 n01910747
n02113799 n02110185 n02672831 n02094433 n03773504 n04147183 n02356798 n02092339
n01514859 n02128385 n04522168 n02823750 n02939185 n01534433 n03495258 n01833805
n02088238 n02951358 n01632777 n07720875 n02106166 n02701002 n01677366 n01644373
n01748264 n02113624 n02437616 n02226429 n01614925 n03467068 n09472597 n03930630
n02219486 n02091032 n02109525 n07920052 n03481172 n02129165 n02106030 n04347754
n02841315 n02317335 n03494278 n02445715 n02077923 n02410509 n02112137 n02906734
n07742313 n01784675 n02480495 n02113023 n04118538 n09835506 n01847000 n04552348
n12267677 n04536866 n02607072 n02423022 n02481823 n01443537 n02486410 n02091134

'goldfish', 'great white shark', 'hammerhead', 'stingray', 'hen', 'ostrich', 'goldfinch', 'junco', 'bald eagle', 'vulture', 'newt', 'axolotl', 'tree frog', 'iguana', 'African chameleon', 'cobra', 'scorpion', 'tarantula', 'centipede', 'peacock', 'lorikeet', 'hummingbird','toucan','duck', 'goose', 'black swan', 'koala', 'jellyfish', 'snail', 'lobster', 'hermit crab', 'flamingo', 'american egret', 'pelican', 'king penguin', 'grey whale', 'killer whale', 'sea lion', 'chihuahua', 'shih tzu', 'afghan hound', 'basset hound', 'beagle', 'bloodhound', 'italian greyhound', 'whippet', 'weimaraner', 'yorkshire terrier', 'boston terrier', 'scottish terrier', 'west highland white terrier', 'golden retriever', 'labrador retriever', 'cocker spaniels', 'collie', 'border collie', 'rottweiler', 'german shepherd dog', 'boxer', 'french bulldog', 'saint bernard', 'husky', 'dalmatian', 'pug', 'pomeranian', 'chow chow', 'pembroke welsh corgi', 'toy poodle', 'standard poodle', 'timber wolf', 'hyena', 'red fox', 'tabby cat', 'leopard', 'snow leopard', 'lion', 'tiger', 'cheetah', 'polar bear', 'meerkat', 'ladybug', 'fly', 'bee', 'ant', 'grasshopper', 'cockroach', 'mantis', 'dragonfly', 'monarch butterfly', 'starfish', 'wood rabbit', 'porcupine', 'fox squirrel', 'beaver', 'guinea pig', 'zebra', 'pig', 'hippopotamus', 'bison', 'gazelle', 'llama', 'skunk', 'badger', 'orangutan', 'gorilla', 'chimpanzee', 'gibbon', 'baboon', 'panda', 'eel', 'clown fish', 'puffer fish', 'accordion', 'ambulance', 'assault rifle', 'backpack', 'barn', 'wheelbarrow', 'basketball', 'bathtub', 'lighthouse', 'beer glass', 'binoculars', 'birdhouse', 'bow tie', 'broom', 'bucket', 'cauldron', 'candle', 'cannon', 'canoe', 'carousel', 'castle', 'mobile phone', 'cowboy hat', 'electric guitar', 'fire engine', 'flute', 'gasmask', 'grand piano', 'guillotine', 'hammer', 'harmonica', 'harp', 'hatchet', 'jeep', 'joystick', 'lab coat', 'lawn mower', 'lipstick', 'mailbox', 'missile', 'mitten', 'parachute', 'pickup truck', 'pirate ship', 'revolver', 'rugby ball', 'sandal', 'saxophone', 'school bus', 'schooner', 'shield', 'soccer ball', 'space shuttle', 'spider web', 'steam locomotive', 'scarf', 'submarine', 'tank', 'tennis ball', 'tractor', 'trombone', 'vase', 'violin', 'military aircraft', 'wine bottle', 'ice cream', 'bagel', 'pretzel', 'cheeseburger', 'hotdog', 'cabbage', 'broccoli', 'cucumber', 'bell pepper', 'mushroom', 'Granny Smith', 'strawberry', 'lemon', 'pineapple', 'banana', 'pomegranate', 'pizza', 'burrito', 'espresso', 'volcano', 'baseball player', 'scuba diver', 'acorn'.

**ImageNet-200-C** (Hendrycks & Dietterich, 2018) is derived from `ImageNet-200` and consists of multiple algorithmically generated corruptions from noise, blur, weather, and digital categories.

## G  THE USE OF LARGE LANGUAGE MODELS (LLMs)

The authors declare that large language models (LLMs) were used solely for polishing the writing of this manuscript. No part of the theoretical development, algorithm design, experimental implementation, data analysis, or other research-related tasks involved the use of LLMs.

