# OpenReview forum: "A Unified and Data-Efficient Framework for Out-of-Distribution and Generalization"
_ICLR.cc/2026/Conference — ICLR 2026 Conference Withdrawn Submission_

### Official Review · Reviewer_WZJc · 2025-10-19

**Soundness:** 2
**Presentation:** 1
**Contribution:** 2
**Rating:** 2
**Confidence:** 4

**Summary:**

The paper claims a unified framework (TODG) that simultaneously tackles OOD detection and OOD generalization via a KL-based “feature regularization” term and implicit OOD feature sampling, i.e., Gaussian.

**Strengths:**

- Clear task framing and a single algorithmic scaffold.
- Ablations suggest each component helps.

**Weaknesses:**

- The core regularizer is defined via KL terms between “semantic-shifted” and “covariate-shifted”, later reformulated by treating one term as a constant $Q$. This assumption is crucial to the argument but is not justified empirically or theoretically.

- the empirical evidence is narrow and often tuned to one specific corruption (“snow”) for ImageNet-200, which does not convincingly establish robustness across distribution families.

- Provide exact fine-tuning schedules, optimizers, and default $\alpha, \beta, \gamma$ values per dataset/setup (not only sensitivity ranges). The “Additional training details” currently emphasize pretraining only.

- The regularizer connecting distributions via KL terms and the Gaussian feature-sampling for augmentation feel close in spirit to prior OE extensions and recent fine-tuning-with-auxiliary-data methods (WOODS, VOS, NPOS).

- Table 1 exists repetitive values across the methods of OOD detection. I am confused with the **authenticity** of the experiments.

**Questions:**

Refer to Weaknesses.

---

### Official Review · Reviewer_R3of · 2025-10-28

**Soundness:** 3
**Presentation:** 2
**Contribution:** 2
**Rating:** 4
**Confidence:** 4

**Summary:**

This paper proposes TODG, a novel method to tackle both ood detection and generalization. TODG designs a regularization term to mitigate the observed Optimization Conflict (OC) issue, while introducing a data generation strategy to alleviate the scarcity of OOD data from both covariate-shifted and semantic-shifted perspective. Experiments on multiple benchmarks validate the superiority of the method.

**Strengths:**

1. The motivation of this paper is clearly stated, and the method is compatible with the proposed issues.
2. The results show the superior performance compared with previous methods.

**Weaknesses:**

1. Limited Technical contribution. The designed loss functions include common CE loss design on the ID data and covariate-shifted data with GT label, as well as a CE loss on the semantic-shifted data with a uniform distribution label, which has been proposed in many previous works to achieve maximal entropy, such as [1]. The OOD data generation pipeline also resembles to existing literatures that use mean and variance for synthesis [2].
[1] Ai W, Yang Z, Chen Z, et al. Maximum open-set entropy optimization via uncertainty measure for universal domain adaptation[J]. Journal of Visual Communication and Image Representation, 2024, 101: 104169.
[2] Li, Xiaotong, et al. "Uncertainty modeling for out-of-distribution generalization." arXiv preprint arXiv:2202.03958 (2022).

2. There’s no clarification on how the x_e^j and x_id^k is obtained during the training process in Equation 25. More training details should be presented on how to obtain these features based on the ID and two types of OOD data.

3. A more detailed training pipeline should be supplemented for better reading on how to train different types of data, including ID, semantic-shifted and covariate-shifted data. Why ID data is used for only 10 epochs, while 100 epochs for semantic-shifted data? How to choose the optimal training epochs if confronting new data distribution?

4. As the KL divergence between the semantic-shifted feature with covariate-shifted is transformed to the KL divergence between the environment-related feature and ID feature, it’ll be better to present a visualization on the value of these two KL values during training to better illustrate the positive correlation.

5. The hyper-parameter analysis is insufficient. For example, why the hyper-parameter alpha and gamma cause huge performance degradation in Figure 3(d) when setting to 7.5 and 10? And why the optimal beta, gamma is set to 1, which contradicts with Figure 3(c). It seems there’s contradictory between your optimal choice and the hyper-parameter analysis.

**Questions:**

Please refer to weaknesses.

---

### Official Review · Reviewer_nphr · 2025-10-28

**Soundness:** 2
**Presentation:** 2
**Contribution:** 2
**Rating:** 2
**Confidence:** 5

**Summary:**

This paper proposes a unified framework to simultaneously address two critical challenges, OOD detection which handling semantic shifts ,and OOD generalization which handling covariate shifts. The authors first provide a theoretical analysis that identifies an "Optimization Conflict" (OC) for environment-related features. To mitigate this conflict, they introduce TODG (Tackling OOD Detection and Generalization) : a feature regularization term and an implicit data generation strategy.

**Strengths:**

The paper tackles an important problem that unify the OOD detection and generalization.

**Weaknesses:**

1. The derivation of Optimization Conflict (OC) in Equation (28) assumes a one-to-one correspondence between class and environment-related components in covariate-shifted OOD data. This may not hold if environmental factors are shared across classes, which would alter the expected behavior of the model.  if the environmental component is shared across all classes, its expected value also seems to be $\frac{1}{c}$.
2. The implicit data generation method assumes that OOD features follow a unimodal Gaussian distribution. This assumption is not sufficiently justified, especially in complex, multi-modal feature spaces, and lacks both theoretical and empirical validation.
3. The paper does not compare its data generation strategy with existing virtual OOD synthesis methods (e.g., VIM, NPOS) in terms of theory or performance, leaving its novelty and effectiveness unclear.
4. The theoretical setup application is narrow: it requires the presence of both covariate-shifted and semantic-shifted OOD data during training, uses a cross-entropy loss combined with outlier exposure, which can not apply to other methods, such as outlier class.

**Questions:**

N/A

---

### Official Review · Reviewer_t2eC · 2025-10-30

**Soundness:** 3
**Presentation:** 3
**Contribution:** 2
**Rating:** 4
**Confidence:** 3

**Summary:**

Tackling OOD Detection and Generalization (TODG) introduces a regularization term to mitigate the optimization conflict that arises when jointly addressing OOD generalization and OOD detection, and it employs a data-generation strategy to alleviate the scarcity of OOD samples. Extensive experiments show that TODG outperforms existing methods, demonstrating strong effectiveness for both OOD detection and generalization.

**Strengths:**

1. The paper introduces a data-efficient framework that jointly tackles semantic-shift OOD detection and covariate-shift generalization, an important need in real-world deployments.

2. This work adopts an explicit optimization-conflict perspective and proposes a principled regularizer to resolve it; the approach is technically sound and well motivated.

3. Extensive experiments across diverse datasets demonstrate clear, consistent improvements over strong baselines.

**Weaknesses:**

1. The paper’s motivation suggests prior work treats OOD generalization and OOD detection separately, which overstates the gap. Several papers already pursue joint treatment [1, 2, 3]. Reposition the contribution with a focused related-work subsection, spell out the precise mathematical/algorithmic differences.

2. Relying on Gaussian feature sampling may be fragile if the true feature distribution is multi-modal or heavy-tailed. Diagnose fit and compare against non-parametric or richer models (k-NN density/score, energy models, or GMMs).

3. Generality to modern encoders is unclear. Evaluate  ViT backbone and CLIP encoder (image/text), and discuss any layer-selection nuances that arise.

4. Stability with respect to regularizer weights, number of sampled features, and OOD/ID minibatch ratios is under-documented.

5. Component ablations and split metrics. Run ablations for each component and report separate metrics for OOD detection (AUROC/FPR95) and OOD generalization (accuracy/robustness).

6. The two objectives can conflict in practice. Analyze conditions where the unified framework degrades (e.g., certain corruption types/severities, semantic classes, or layers). Include qualitative/error analyses and provide guidance for detecting and mitigating these cases.

Reference:

[1] Bridging OOD Detection and Generalization: A Graph-Theoretic View

[2] InfoBound: A Provable Information-Bounds Inspired Framework for Both OoD Generalization and OoD Detection

[3] AHA: Human-Assisted Out-of-Distribution Generalization and Detection

**Questions:**

1. Several recent papers already pursue joint OOD generalization and detection. What are the precise mathematical and algorithmic differences versus TODG? Please clarify these distinctions to better position the contribution and its novelty.

2. Have you tested whether the chosen feature layer is approximately Gaussian? How does performance change when using non-Gaussian alternatives such as KDE or energy-based models?

3. Do the observations and trends hold on modern backbones such as ViT and CLIP encoders, and have you verified that the OOD sets are truly unseen during training/pretraining?

4. Beyond intuition, can you provide empirical diagnostics that quantify the conflict between OOD generalization and OOD detection (e.g., gradient cosine similarity, loss interference), and show how the regularizer reduces this conflict during training?

5. Under what conditions does TODG underperform (e.g., specific corruption types, severity levels, or semantic classes)? Please include analyses or case studies.

---

### Note · Authors · 2026-01-16

I have read and agree with the venue's withdrawal policy on behalf of myself and my co-authors.